# Specific *Mycobacterium tuberculosis* Strain Circulating in Prison Revealed by Cost-Effective Amplicon Sequencing

**DOI:** 10.3390/microorganisms12050999

**Published:** 2024-05-15

**Authors:** Joaquín Hurtado, María Noel Bentancor, Paula Laserra, Cecilia Coitinho, Gonzalo Greif

**Affiliations:** 1Laboratorio de Evolución Experimental de Virus, Institut Pasteur Montevideo (IPM), Montevideo 11400, Uruguay; jhurtado@pasteur.edu.uy; 2Comisión Honoraria de Lucha Anti-Tuberculosa y Enferemedades Prevalentes, Montevideo 11200, Uruguay; manoben2011@gmail.com (M.N.B.); paulalaserra@gmail.com (P.L.); ccoithino80@gmail.com (C.C.); 3Laboratorio de Interacciones Hospedero-Patógeno, Institut Pasteur de Montevideo, Montevideo 11400, Uruguay

**Keywords:** tuberculosis, epidemiology, amplicon sequencing

## Abstract

This scientific study focuses on tuberculosis (TB) within prison settings, where persons deprived of liberty (PDL) face significantly higher rates of the disease compared to the general population. The research employs the low-cost amplicon sequencing of *Mycobacterium tuberculosis* strains, aiming first to identify specific lineages and also to detect mutations associated with drug resistance. The method involves multiplex amplification, DNA extraction, and sequencing, providing valuable insights into TB dynamics and resistance-mutation profiles within the prison system at an affordable cost. The study identifies a characteristic lineage (X) circulating among PDLs in the penitentiary system in Uruguay, absent in the general population, and notes its prevalence at prison entry. No high-confidence mutations associated with drug resistance were found. The findings underscore the importance of molecular epidemiology in TB control, emphasizing the potential for intra-prison transmissions and the need for broader studies to understand strain dynamics.

## 1. Introduction

Tuberculosis disease (TB) still represents the major cause of death by an infectious agent (2023 WHO Tuberculosis global report), second only to SARS-CoV-2 in 2020 and 2021. Every year, approximately 10 million people fall ill with TB and despite being a preventable and curable disease, 1.5 million people die (WHO report 2023).

The incidence of TB varies among regions, from high-burden regions (234/100,000 inhabitants in South-East Asia regions), to low-incidence regions (25/100,000 inhabitants in European regions). The WHO Tuberculosis Global Report 2023 indicates an incidence of 31/100,000 inhabitants for American regions. In 2021 and 2022, the increased number of cases reverted the global reduction rate observed between 2010 to 2019, due to the impact of COVID-19.

The Latin America data indicates rates that range from high-incidence (151/100,000 inhabitants in Perú) to very-low-incidence rates (6.6/100,000 inhabitants in Cuba). Uruguay reported an incidence of 38/100,000 inhabitants in 2022, showing a continue increase since 2010 (opposite to the global decrease trend) (WHO Report 2023). The phenomenon was influenced by the increase in populations at very high risk linked to social issues: incarcerated individuals, residents of marginalized areas, and patients with AIDS.

The tuberculosis situation is particularly dire within prison settings, where persons deprived of liberty (PDL, following the Inter-American Commission on Human Rights) experience alarmingly high rates of the disease. Globally, the incidence of TB is tenfold higher among incarcerated individuals compared to the general population. Latin America, in particular, faces an acute challenge, with the TB incidence in prisons being 26.9 times greater than that in the general population [1]. In terms of global incidence rates, there is a decreasing trend in the incidence of TB in prison, except in the case of Central and South America, where, since 2000, the incarcerated population has grown by 206%, the highest increase in the world. Over the same period, notified tuberculosis cases among PDLs have risen by 269% [2].

While PDLs are recognized by the World Health Organization (WHO) as a high-risk population for tuberculosis, global data on tuberculosis cases among PDLs are not systematically gathered [3]. In Latin America, reports on tuberculosis in prisons are scarce, with a majority of these from Brazil [4,5,6,7,8,9,10,11]. In Uruguay, where this study was performed, the available data indicate an incidence rate of 887/100,000 PDL (23.3 times greater than in the general population, in line with the reported data [2]) (https://chlaep.org.uy/programa-nacional-de-control-de-tuberculosis/dia-de-la-tuberculosis-2023/ accessed on 4 May 2024). The number of cases and the resistance bacterial profile is the only recorded information in the country.

In an effort to gain insight into the prevalence of *Mycobacterium tuberculosis* strains within prisons in Uruguay, we opted to conduct low-cost amplicon sequencing of specific regions of the *M. tuberculosis* genome. For this study, all cases detected in prison in 2019 were included (*n* = 47). The majority of the cases were detected upon entry to the penitentiary system (28/47) and were initially housed in Penitentiary prison N°4. Located in the capital of the country, where approximately 50% of the population lives, Prison N°4 houses 30% of the total PDLs in the country (in total, 32/47 strains in this study were collected from here, including the cases detected at the entry of the system). Seven strains (7/47) were detected at Penitentiary prison N°6 (with approximately 5% of the PDL total population, located on the other side of the country’s capital) and 4/47 were isolated from Penitentiary prison N°7 (8% of total PDL population) located in the second most populated region of the country. The remaining four cases correspond to female patients deprived of liberty in two different detention centers (one in the capital—Penitentiary N°5—and three in the center of the country—Penitentiary N°18).

The developed method not only identifies the lineage but also detects mutations in genes associated with resistance to first- and second-line drugs used in the treatment of tuberculosis (TB). The SNP-based approach allows us to amplify eleven regions containing characteristic lineage SNPs and nine genes associated with anti-tuberculosis drugs using two simple PCR pool reactions, at an affordable cost for the public system. With this approach, we identified a specific lineage (lineage X) circulating among PDL that is not present in the general population [11]. Interestingly, all the cases detected upon entry into the prison system with the X lineage had a previous history of incarceration. The information about this lineage is particular scarce, but was identified as having caused outbreaks with a high transmission rate in the USA in 1995 and in Spain between 1990 and 2018 [12].

## 2. Materials and Methods

### 2.1. Samples

The genomic DNA extraction was carried out within the framework of the project FMV_3_2018_1_148367: “Generation of an economic system for molecular typing of *M. tuberculosis* strains and its implementation in strains isolated in the country.” The Bacteriological Laboratory of CHLA-EP centralizes all tuberculosis bacteriology in the country and performs cultures in a Löwestein–Jensen medium from various biological samples (sputum, tracheal aspirate, bronchoalveolar lavages, etc.). Genomic DNA was extracted from positive cultures using standardized methods with the Genolyse kit (HAIN, Nehren, Germany) and stored at −20 °C in the Biobank of CHLA-EP.

A total of 47 DNA samples were obtained from isolates from individuals in the prison system diagnosed with tuberculosis during the year 2019 (Appendix A), representing all isolates of the year. The diagnosis of pulmonary TB was made based on the evidence of clinical and radiological symptoms, followed by confirmation through bacterial culture.

### 2.2. Multiplex Amplification

The lineage determination was conducted through a search for single-nucleotide polymorphisms (SNPs) to classify lineages (Appendix A). These SNPs are reported in the literature and enable the discrimination of known circulating strains [13]. Additionally, the primer design was carried out to determine the sequence of genes where mutations associated with resistance to first- and second-line anti-tuberculosis drugs were reported (Appendix A).

For the primer design, web-based tools, such as PrimerPooler v1.88 [14], were utilized. Subsequently, the formation of dimers and heterodimers was verified, taking into account the possibility of performing multiple amplifications in a single reaction, using online tools available at “Integrated DNA Technologies” (www.idt-dna.com accessed on 5 January 2020).

The validation of all amplicons was performed independently, followed by the determination of all necessary parameters for simultaneous amplification. Finally, the concentration of each primer was optimized to form the lineage pool (pool-L) and the resistance pool (pool-R). The lists of primers and amplified regions are summarized in Appendix A.

### 2.3. Library Construction and Sequencing

Barcoded libraries compatible with the MiSeq platform (Illumina, San Diego, CA, USA) were constructed using the “Nextera XT DNA library Prep” kit (Illumina) following the manufacturer’s instructions. A 2 × 75 (paired end) sequencing kit was used at the Illumina MiSeq equipment, available at the Institut Pasteur de Montevideo.

### 2.4. Bioinformatics Analysis

We developed our own analysis scheme, concatenating command-line programs to identify single-nucleotide polymorphisms (SNPs) for lineage and resistance classification. First, a check of the quality of the output files or reads in fastQ format was performed, trimming the ends with qualities lower than 30; adapter sequences were removed using Sickle v1.33 [15]. Quality trimmed reads were mapped against H37Rv *M. tuberculosis* reference genome using the bwa algorithm [16]. Samtools v1.13/BCFTools v1.18 [17,18] was used to identify the SNPs. Finally, each SNP was manually validated visualizing the mapping file in IGV v2.8.0 [19] software. The complete pipeline is included in Appendix A. Additionally, validation was performed by comparing results with the results of phyresse [20] pipeline. To validate the lineage results, we selected samples and performed MIRU-VNTR [21] and spoligotyping [22]. Information about coverage by amplicons in resistance and lineage regions are summarized in Appendix A.

## 3. Results

### 3.1. Circulating Lineages among PDL

We first evaluated the coverage of the amplicon-covered region for lineage determination. We obtained values ranging from 23 to 14,413 (with a mean of 1200). Only one sample was discarded because we could not obtain sequencing reads by performing either MIRU-VNTR or Spoligotyping, suggesting the presence of inhibitors in the sample. The genotyping results are summarized in Table 1.

The results obtained indicate that, within the PDL population, 39% of strains were identified as LAM (lineage 4.3), 34% as X (lineage 4.1.1), 25% as Haarlem (lineage 4.1.2.1), and 2% as S (lineage 4.4.1.1) (Figure 1). The distribution of lineages contrasts with that observed in community isolates (data obtained from previous work [11]). In particular, the X strain is overrepresented in PDL (chi-square with *p*-value = 0.0001). This lineage was not detected in any isolates from patients admitted to intensive care units and is detected in very low percentages in community isolates (around 1%) [11]. It was also not observed in drug-resistant strains previously characterized by our group [23].

### 3.2. Drug Resistant Profile

For the search of mutations associated with drug resistance, the same work scheme in the previous section was used, narrowing the search to the selected drug-resistance-associated genes. Table 2 summarizes all the polymorphisms detected. Only in four isolates, reads could not be obtained, and the analysis for the search for resistance polymorphisms could not be performed.

Most mutations are synonymous without alterations in an amino acid sequence (57%). In three isolates, the c1472337t mutation was detected in the *rrs* gene, which codes for the 16S ribosomal subunit. The mutation is associated with streptomycin resistance, although it is not characterized as a high-confidence SNP [25]. In the case of the other non-synonymous mutations, no bibliographic evidence of phenotypic resistance was found and the results of the sensitivity studies on these strains indicate that these mutations were not associated with resistance.

## 4. Discussion

Through this work, primers were successfully designed to amplify specific regions of the *M. tuberculosis* genome in a multiplex format. Two independent multiplexes were designed: one amplifying regions with characteristic polymorphisms of different lineages and sub-lineages within the species, and the other targeting nine genes associated with resistance to major anti-tuberculosis drugs. These nine genes were fully amplified, covering not only regions with known mutations but the entire coding region.

To obtain lineage and resistance data for each strain, the amplification of each of these multiplexes (referred to as Lineage pool and Resistance pool) was performed using Illumina technology.

Since the selection of SNPs to determine lineages was performed in 2018, new sequencing data and analysis have provided more information about a robust set of SNPs capable of typing *M. tuberculosis* strains [26,27]. However, all the SNPs used in this study to determine lineage were included as representative of lineages in [26], except for the SNP used for lineage 6 (not relevant in this study). In light of the results, primer design could be improved by adding new SNPs per lineage. It could also be adapted for specific regions. For example, in Uruguay, there were only lineage 4 is circulating; thus, only SNPs related to this particular lineage could be included in the multiplex to delve deeper and have more confidence in the lineage determination.

Libraries were sequenced in paired-end runs of 75 bases to achieve a coverage of 100× for the total amplified bases (approximately 20,000 bases); approximately 15,000 sequences per sample were required. Although the library production costs were the same whether obtaining the entire genome or only selected regions, the sequencing of complete genomes incurs higher costs. However, the information derived from complete genomes allows for more precise lineage determination, better outbreak monitoring, the identification of reinfection or relapse cases, and even the discovery of mutations not yet associated with drug resistance. As sequencing costs decrease and accessibility improves, the routine sequencing of complete genomes may become more feasible in the future. For now, the alternative proposed in this project offers the possibility of obtaining relevant information at a lower cost with similar analysis times, presenting itself as a viable option for local implementation, whether for all isolates or those of greater interest (vulnerable populations, outbreaks, drug-resistant strains, etc.).

With regard to the capacity to obtain lineage information using this strategy, we obtained 100% agreement with samples typed with MIRU-VNTR/Spoligotyping (Table 1). In addition, our analysis pipeline was validated with 100% agreement using amplicon data from the Phyresse online tool or/and WGS data for selected strains (Table 1).

According to the typing analysis conducted in this proof-of-concept study, the X strain was found in more than 30% of isolates from PDL in 2019, while its proportion is around 1% in the community isolated strains. When examining strains, we found that at the initial point of entry into the prison system (PS) (when most TB cases are detected), the percentage of X strains is higher than 30%. The examination showed that, in particular, only PDLs with previous stays in the PS had this strain. The first-time entry of PDLs into the PS carried strains other than X.

These findings suggest the possibility of intra-PS transmissions and low dissemination in the community. However, expanding the study to include family members, police, and other system actors could reveal more information about the dynamics of this strain. This discovery and its prospective analysis highlight the importance of molecular epidemiology in tuberculosis and its role in disease control, particularly in the development of specific policies, monitoring, and population-level impact assessments.

Institutions, such as hospitals and prisons, not only act as amplifiers of TB in the community but also increase the risk of multidrug-resistant tuberculosis (MDR-TB) occurrence [28]. In this work, mutations were detected in the majority of isolates from PDLs; however, 57% of these mutations are synonymous and are not reported as resistant to antibiotics. For the non-synonymous mutations, at least 27 isolates were found to have mutations in the *gyrA* gene: Glu62Gln, Ser95Thr, and Gly668Asp. This gene is associated with resistance to fluoroquinolones, and some mutations are known for their high confidence in phenotypic resistance. For example, the Asp94Gly mutation is frequently observed in extensively drug-resistant (XDR) strains [29]. The mutations found in the study population have not been reported as having an association with drug resistance.

Additionally, two-point mutations were found in the *rpoB* gene in the following two different isolates: Leu275Val in 100% of the sequences (163× coverage) and the Met349Thr mutation in 99% of the sequences (169× coverage). However, these mutations are not reported as being associated with phenotypic resistance, nor are they located in the hotspot region where most resistance-causing mutations in this gene occur.

While our study did not identify mutations associated with resistance, it is crucial to emphasize the significance of such investigations, especially at the initiation of treatments. Detecting resistance mutations early in the treatment process can guide clinicians in adapting therapeutic strategies, ensuring more effective patient care. Furthermore, the determination of lineages is pivotal for conducting thorough epidemiological studies and implementing proactive measures for outbreak control. Understanding the genetic characteristics of *M. tuberculosis* strains within prison settings not only aids in individualized patient management but also contributes to the broader efforts of disease surveillance, intervention planning, and preventing potential outbreaks.

## Figures and Tables

**Figure 1 microorganisms-12-00999-f001:**
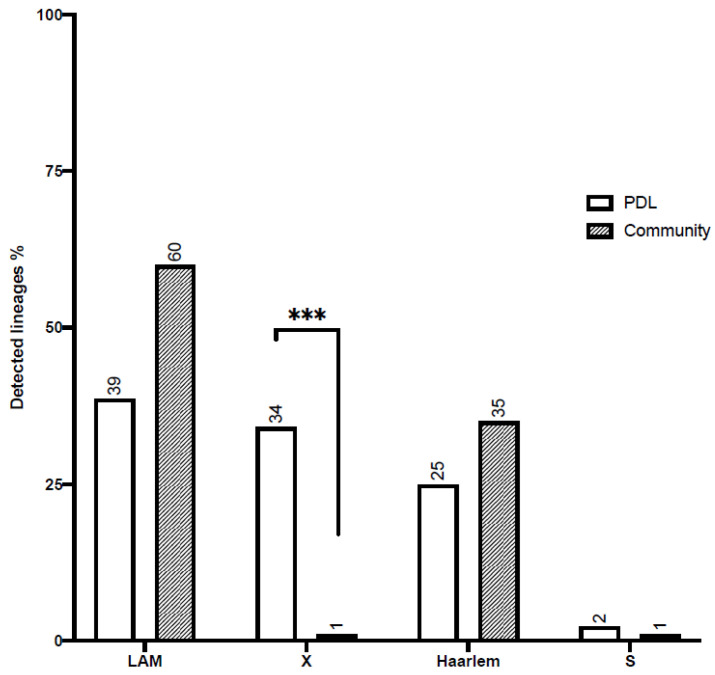
Lineage distribution among PDL and community *M. tuberculosis* isolates. The bar plot shows the percentage of detected lineages among PDL (white) and community isolates (gray) *** indicate a chi-square test result lower than 0.0001.

**Table 1 microorganisms-12-00999-t001:** Results of lineage determination. * Cases performed to validate the method. ** Genotype assigned by MIRU-VNTR.

Sample ID	Coverage Pool-L	Genotype SNP	Genotype Phyresse/MIRU-VNTR *
1	104	LAM	LAM
2	1713	LAM	LAM
3	317	X	X
4	188	LAM	LAM
5	348	Haarlem	
6	96	X	X
7	8081	X	X
8	-	-	
9	2103	Haarlem	
10	187	X	X
11	3470	LAM	LAM
12	203	LAM **	/LAM
13	303	Haarlem	
14	399	LAM **	/LAM
15	2051	X	X
16	143	Haarlem	
17	413	Haarlem	
18	281	X	
19	1984	Haarlem	
20	281	X	
21	187	Haarlem	
22	360	S	S
23	1725	LAM	LAM
24	771	X	
25	176	LAM	LAM
26	367	X	X/Haarlem
27	6437	-	
28	14,413	Haarlem	Haarlem
29	178	X	
30	915	X	
31	557	LAM	LAM LAM
32	444	LAM	
33	153	LAM **	/LAM
34	526	LAM **	/LAM
35	271	LAM	LAM
36	346	Haarlem	
37	82	X	
38	23	X	X
39	1559	LAM	LAM
40	555	X	
41	88	-	
42	382	Haarlem	
43	234	LAM	LAM
44	310	LAM	LAM
45	167	LAM	LAM
46	99	X	
47	1200	X	X

**Table 2 microorganisms-12-00999-t002:** Results of detected mutations in genes associated with drug resistance. Bold: missense mutations. * Not associated with phenotypic resistance.

Gene	Genome Position	Samples (n)	SNP	Codon Change	Amino Acid Change	Previous Reported
*gyrA*	7362	30	G/C	GAG/CAG	Glu62Gln	Not reported
*gyrA*	7585	28	G/C	AGC/ACC	**Ser95Thr**	Reported [24] *
*gyrA*	8040	4	G/A	GGC/AGC	**Gly247Ser**	Reported [24] *
*gyrA*	9304	27	G/A	CGC/GAC	**Gly668Asp**	Reported [24] *
*gyrB*	6438	1	C/G	CCC/CGC	**Pro400Arg**	Not reported
*gyrB*	6034	1	C/T	AAC/AAT	Asn265Asn	Not reported
*katG*	2,154,279	2	G/A	CTC/CTT	Leu611Leu	Not reported
*katG*	2,154,015	1	C/G	GGG/GGC	Gly699Gly	Not reported
*pncA*	2,289,017	17	A/G	GGT/GGC	Gly75Gly	Not reported
*rpoB*	760,106	2	G/A	TCG/TCA	Ser100Ser	Not reported
*rpoB*	760,115	10	C/T	GAC/GAT	Asp103Asp	Not reported
*rpoB*	760,629	1	C/G	CTG/GTG	**Leu275Val**	Not reported
*rpoB*	760,852	1	T/C	ATG/ACG	**Met349Thr**	Not reported
*rrs*	1,472,337	3	C/T	Ribosomal	c492t	Streptomicin [25]
*rrs*	1,473,314	4	A/C	ribosomal	a1549c	Not reported

## Data Availability

The raw data supporting the conclusions of this article will be made available by the authors on request.

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
