# Peer review of "Specific Mycobacterium tuberculosis Strain Circulating in Prison Revealed by Cost-Effective Amplicon Sequencing"

_microorganisms, 2024, doi:10.3390/microorganisms12050999_

Round 1

Reviewer 1 Report

Comments and Suggestions for Authors

(1) ABSTRACT

Abstract: “ individuals deprived of liberty (PDL)”  - actually PDL cannot be abbreviated in this way. Do you mean persons deprived of liberty?

Abstract – be more specific, write where the study was conducted. As it stands this sounds global and general. This lineage X is apparently specific for your particular setting.

General comment on abstract – “ aiming to identify specific lineages and mutations associated with drug resistance”

Resistance mutations (if not very specific and rare) cannot serve as markers of transmission. Even lineage markers should be of high resolution to trace reliable transmission.

Overall, the abstract is formulated in very general and hard to understand way (indeed 200 words limit may be a reason)

(2) INTRODUCTION

Again, in the last para of introduction, write about your location.

(3) METHODS.

Please – do write where did you perform this study? Country, city? Name of prison.

(4) Regarding choice of SNPs, you refer to the first (but outdated paper) of Coll et al. 2014.

There were more recent and critical updates.

A study by Coll et al. in 2014, was updated by Napier et al. in 2020 and summarised in Shitikov, Bespiatykh 2023.

I strongly recommend to read these papers and Refs therein, to see if your SNPs are still valid.

Napier G, Campino S, Merid Y, Abebe M, Woldeamanuel Y, Aseffa A, Hibberd ML, Phelan J, Clark TG. 2020. Robust barcoding and identification of Mycobacterium tuberculosis lineages for epidemiological and clinical studies. Genome Med 12:114.

Shitikov E, Bespiatykh D. A revised SNP-based barcoding scheme for typing Mycobacterium tuberculosis complex isolates. mSphere. 2023 Aug 24;8(4):e0016923. doi: 10.1128/msphere.00169-23.

(5) “The strain typing was conducted through the search for single nucleotide polymorphisms (SNPs) to classify lineages” – this approach is very low-resolution. Knowledge about lineages is not enough to trace transmission. This approach can hardly be termed as strain typing.

(6) “Additionally, primer design was carried out to determine the sequence of genes where mutations associated with resistance to first and second-line anti-tuberculosis drugs were reported (Supplementary Table 3).”

As you know, only parts of these resistance genes concern resistance; otherwise, their mutations are usually phylogenetic markers

Regarding the role of resistance mutations – see the 2nd edition of the WHO catalogue published 2023.

(7) DISCUSSION

“the other targeting nine genes associated with resistance to major anti-tuberculosis drugs. These nine genes were fully  amplified”

Their full amplification is just unnecessary. – see WHO catalogue

On the other hand as I wrote – lineage determination has nothing to do with real strain typing that must be high-resolution.

(8) Where did this X lineage come from to your prison setting? If not from general population. If you had WGS data you could perhaps estimate time of its origin?

(9) I also wonder how cost effective can be sequencing of so many genes (full length genes!).

When you write about “low-cost amplicon sequencing of spe-60 cific regions of the M. tuberculosis genome.”

– did you calculate the cost?

(10) other comments

Mycobacterium tuberculosis – in italic, as well gene names – check and correct in several places

In Abstract. Mycobacterium tuberculosis – on first use, M. tuberculosis – on all further use.

In the main text: the same. Mycobacterium tuberculosis – on first use, M. tuberculosis – on all further use. E.g. at the end of Discussion you write it in full, which is wrong.

Line 147 - Not rrS gene but rrs (in italic)

Keywords: you did not study “pathogenesis”.

Author Response

My response in red in the Word file. Thanks

Reviewer 2 Report

Comments and Suggestions for Authors

Major comment:

-         - The authors neglect to discuss the most important part of the paper: how exactly the pool-L was used to identify the lineage, and the level of agreement between their amplicon method and the standard methodology (Phyresee/MIRU-VNTR). Regarding the equivalence, even if a program was not used, the authors need to add to the methods section what algorithm or steps they used.

-         - The coverage and number of reads should be reported by locus amplified. When multiplexing, it is common to see some loci having good results and some with poor results. Descriptive statistics on the average, median, minimum and standard deviation among all samples should be given.   

Minor comment:

-         On line 61, please change “The developed method not only identify”, to “The developed method not only identifies”

-          On table 2, the authors state: ”drug resistant mutations”, but I think they are simply mutations in genes where mutations associated with resistance to drugs have been reported. You don’t know if they are truly drug-resistant, especially difficult to claim for the ones that produce synonymous changes.

-          On line 151, the authors state about the changes not reported in literature: “and it is consistent with information from sensitivity studies previously conducted and with the clinical-bacteriological response observed in the PDL”. I fail to understand what the authors are exactly trying to say here, is it that these are likely not important?

Author Response

My response in red in the Word file.

Reviewer 3 Report

Comments and Suggestions for Authors

The authors are to be applauded for their exploration of this captivating study, showcasing a significant concern for the well-being of individuals residing in prisons and identifying a potentially dangerous strain of Mycobacterium tuberculosis. Their commitment to humanitarian issues is admirable, and their diligent efforts merit recognition. However, there are several key areas that require attention to improve the manuscript's quality for readers.

1. The data presented is from the WHO Tuberculosis Report 2023, which covers data until 2022. It is advisable for the authors to include 2023 data if available. Furthermore, while the authors mention that tuberculosis was "second only to SARS-CoV-2 in 2020 and 2021," it is essential to note that this trend continued in 2022 as well.

2. Line 13: The term "individuals deprived of liberty (PDL)" is introduced for the first time; therefore, it would be more appropriate to use the term "People deprived of liberty."

3. The authors discuss the increasing TB rate in Peru since 2010. It would be valuable to include possible reasons for this trend in the introduction section.

4. The introduction section would benefit from better organization, as it currently appears somewhat scattered.

5. The manuscript primarily focuses on strain "X." It would be beneficial to provide more information about this strain in the introduction section, highlighting its significance, differences from other circulating strains, and its prevalence among the PDL population. Additionally, a brief introduction to other strains discussed in the manuscript would aid readers' understanding.

6. Line 179: Clarification is needed regarding "the initial point of entry into the prison system." Does this refer to when a prisoner first enters the prison? If so, it implies that they were part of the community recently, which may contradict the study's observations.

7. Line 146: Is it meant to refer to a synonymous amino acid change or a synonymous mutation that does not alter the amino acid sequence?

8.  "Inhab" should be spelled out in full, as it is not an acronym.

These revisions will contribute to the manuscript's clarity and improve its overall impact on readers.

Comments on the Quality of English Language

The english seems fine, minor changes are required.

Author Response

My response in red in the Word file.
